# Healthy Parent Carers programme: mixed methods process evaluation and refinement of a health promotion intervention

Jenny Lloyd [1,2] Gretchen Bjornstad,[1,3] Aleksandra Borek [4]
Beth Cuffe-Fuller,[3] Mary Fredlund,[1,3] Annabel McDonald,[3] Mark Tarrant,[1]
Vashti Berry,[1] Kath Wilkinson,[3] Siobhan Mitchell,[1] Annette Gillett,[3] Stuart Logan,[1,3]
Christopher Morris [1,3]

For numbered affiliations see end of article.

**Correspondence to**
Dr Jenny Lloyd;
J.J.Lloyd@exeter.ac.uk

## ABSTRACT

**Objectives** Parent carers of children with special educational needs or disability are at risk of poorer mental and physical health. In response to these needs, we codeveloped the 'Healthy Parent Carers' (HPC) programme. This study examined the views and experiences of participants in the HPC feasibility trial to inform programme refinement.

**Intervention, setting and participants** HPC is a peer-led group-based intervention (supported by online materials) for primary carers of disabled children, encouraging behaviours linked with health and well-being. It was delivered by two lead and six assistant peer facilitators in six community sites (one lead and one assistant per group) in South West England over six or 12 sessions. Control participants had online materials only. The trial involved 47 intervention and 45 control parent carers (97% female and 97% white) and eight facilitators (one male).

**Design** A preplanned mixed methods process evaluation using questionnaires and checklists (during and after the intervention), qualitative interviews with participants after intervention (n=18) and a focus group with facilitators after trial.

**Results** HPC was highly acceptable to participants and facilitators and experiences were very positive. Participants reported that the programme increased awareness of what parent carers could and could not change and their self-efficacy to engage in health-promoting behaviours. The intended mechanisms of action (social identification and peer support) matched participants' expectations and experiences. Control participants found the online-only programme flexible but isolating, as there were no opportunities to share ideas and problem solve with peers, the key function of the programme. Areas for improvement were identified for programme content, facilitator training and delivery.

**Conclusion** HPC was acceptable, well received and offers considerable potential to improve the health of parent carers. Under the pandemic, the challenge going forward is how best to maintain reach and fidelity to function while delivering a more virtual programme.

**Trial registration number** ISRCTN15144652.

## STRENGTHS AND LIMITATIONS OF THIS STUDY

⇒ The Healthy Parent Carers programme has followed the key principles of intervention development.
⇒ Qualitative and quantitative data have been synthesised systematically to refine and optimise the intervention.
⇒ The intervention refinement process is transparent, and adaptations reflect the process data collected.
⇒ Key uncertainties to be addressed in future research have been identified.
⇒ Experiences and views are from a predominantly white population, therefore are not necessarily representative of ethnically/culturally and linguistically diverse parent carers.

## INTRODUCTION

Parent carers of children with special educational needs or disability are at increased risk of poorer mental[1–10] and physical health,[2 3 6 7 11–13] a problem recognised in the NHS Long Term Plan as requiring action to support the personal needs of carers.[14] Parent carers experience challenges to maintaining good health that have implications for their well-being and their ability to care for their children, and recent reviews conclude that there are insufficient programmes that aim to support parental health, which are likely to be the best strategy to advance both child and family outcomes.[15 16] In response to this need we codeveloped the 'Healthy Parent Carers' (HPC) programme, a community-based behaviour change approach to improve health and well-being, advocated by Public Health England.[17 18] This health promotion intervention targets specific behaviours based around a set of universal and evidence-based actions (called CLANGERS) associated with health and well-being. CLANGERS stands for Connect, Learn, be Active, take Notice,

Box 1  Summary of the Healthy Parent Carers programme (including the online materials)

**Face-to-face sessions**
*Content*—activities based on CLANGERS (Connect, Learn, be Active, take Notice, Give, Eat well, Relax and Sleep), an extension of the '5 ways to wellbeing'.
*Format*—12 modules over 24 hours.
*Setting*—community sites (two special schools, one children's hospice, one Parent Carer Forum premise, one adult learning community venue and one hotel regularly used for parent carer meetings).
*Delivery*—six 4-hour daytime sessions (comprising two modules per session) or twelve 2-hour evening sessions (one module per session) delivered to groups of 4–12 parent carers.
*Personnel*—one lead and one assistant facilitator per group.

**Online materials**
Included written content which provided participants with information on the CLANGERS, note-taking space to reflect on their own thoughts and templates for participants to develop their own goals and action plans. Audio and video recordings were also provided to illustrate each of the CLANGERS.
The content related to each module was released to participants in each group after it was delivered in their specific group sessions.

**Box 2  Facilitator training programme**

**Format**
⇒ *Block 1*: 2 days (lead facilitators only). November 2018.
⇒ *Block 2*: 2 days (lead and assistant facilitators). November 2018 (3 weeks following block 1).
⇒ *Block 3*: 1 day (lead and assistant facilitators). April 2018 (after delivery of cohort 1).

**Content**
⇒ *Block 1*. Overview of programme; exploration of well-being and the CLANGERS (Connect, Learn, be Active, take Notice, Give, Eat well, Relax and Sleep); facilitator roles and responsibilities; modelling delivery; research processes; safeguarding.
⇒ *Block 2*. Overview of programme; exploration of well-being and the CLANGERS; facilitator roles and responsibilities; facilitator skills and competencies; group facilitation; managing group dynamics; modelling delivery; practising delivery; research processes; safeguarding.
⇒ *Block 3*. Refresher training; review of CLANGERS, facilitator reflections, research processes, safeguarding.

**Personnel**
Researchers and two parent carers (MF and AM) who coproduced the programme and delivered the proof-of-concept pilot.

**Facilitator recruitment**
Parent carers who are senior facilitators of the Council for Disabled Children's Expert Parent Programme were referred to us to become lead facilitators, based on their experience in developing and facilitating programmes for parent carers.
Assistant facilitators were recruited through adverts shared through contacts in the project Stakeholder Advisory Group. Adverts included information about the role and person specification criteria. Applicants were interviewed by a researcher and selection decisions were made by the research team.

**Safeguarding**
The safeguarding protocol for facilitators was outlined in training and provided in the delivery manual. If any concerns arose during the programme, facilitators were instructed to inform the study team and follow the protocol, which included a reporting flow chart and information for Multi-Agency Referral Units and other relevant contacts at each site.

Give, Eat well, Relax and Sleep.[19] The 'CLANG' component comprises the 'Five Ways to Wellbeing' based on the evidence from the Foresight Project on Mental Capital and Wellbeing.[20] Each of these behaviours is potentially more difficult for parent carers to sustain because of the demands and disruptions of their caring role. The programme involves a range of activities to improve parent carer confidence, motivation and self-efficacy to plan, prioritise and enact these universal actions to improve their own health and well-being, while expanding their social network and providing peer-to-peer social support.

Intervention mapping[21] with extensive stakeholder involvement was used to develop programme content and delivery strategies (online supplemental document 1a,b) which was piloted with one group of seven parent carers, delivered by two peer facilitators (MF/AM), with whom the intervention was cocreated. The findings of this proof-of-concept study and details of the intervention development, logic model and content were published previously.[22] Box 1 summarises the programme content, format and delivery.

### The feasibility study

The feasibility study aimed to assess whether the programme could be delivered in the community and evaluated the acceptability of a randomised controlled trial (RCT) design.[23] The trial ran between July 2018 and June 2020. Ninety-two participants were randomised: 47 to the HPC group programme (delivered in six community sites across Cornwall, Devon and Somerset) and 45 to the control group, which involved access to the HPC online resources only. The group sessions ran between January and July 2019. Outcome measures were collected at baseline (prior to randomisation), immediately after

intervention and 6 months later (online supplemental document 2).

HPC group sessions were delivered by pairs of peer lead and assistant facilitators who were themselves parent carers of children with chronic health conditions. Two lead facilitators, experienced in delivering group training to parent carers, were recruited from the Council for Disabled Children (CDC). Nine assistant facilitators (including three reserves) were recruited through recommendations from our Stakeholder Advisory Group and through local network adverts. Volunteers were shortlisted and interviewed by telephone. Selection decisions were made based on a practical understanding of the challenges faced by parent carers in relation to their own health and well-being with reference to our person specification, and availability to deliver at one of the selected venues. Details of the training programme are shown in box 2.

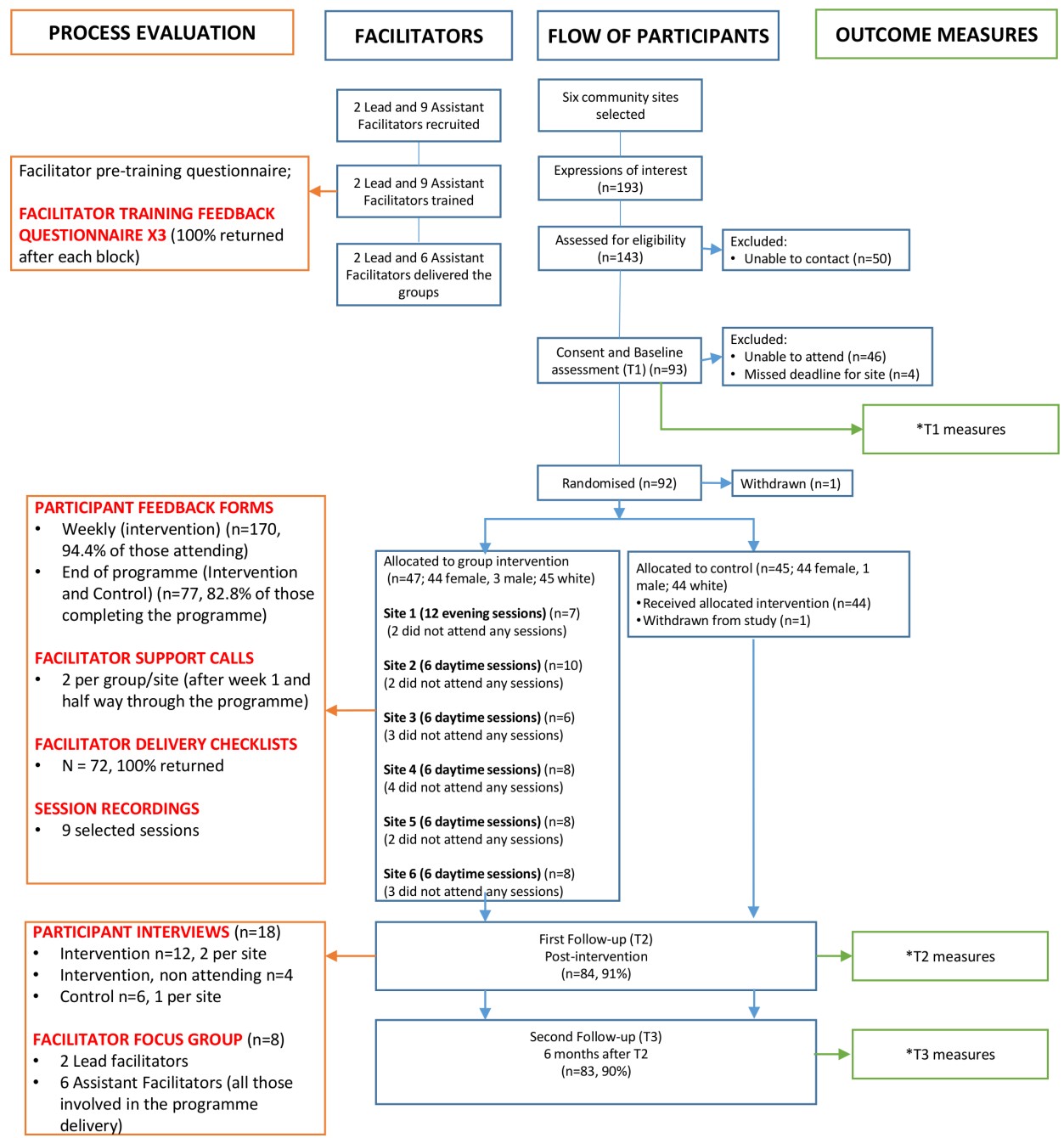

**Figure 1** Flow of participants and measures. *Bjornstad *et al*.[23]

One of the lead facilitators delivered the 6-week programme to two groups (two sites) and one delivered to four groups (four sites), one of which followed the 12-session format (box 1). Each assistant facilitator was assigned to support delivery in one of the six groups.

Figure 1 shows the flow of participants in the trial and details of data collection.

The aim of this paper is to present the mixed methods process data on the motivations of parent carer participants and facilitators to take part in the HPC programme, their views and experiences of receiving/delivering the programme and its refinement in response to these data. Trial findings related to recruitment and retention, fidelity of intervention delivery, the feasibility and

acceptability of trial processes and outcome data are reported separately.[24]

## METHODS
### Patient and public involvement

Parent carers have been involved in all stages of developing the intervention, designing the feasibility study and refining the programme in light of the findings from this feasibility trial, through the involvement of the Peninsula Childhood Disability Research Unit (PenCRU) Family Faculty (www.pencru.org/getinvolved/ourfamilyfaculty). The Family Faculty are parent carers who are offered opportunities to be involved in research.

## Box 3 Process evaluation measures and sampling

**Facilitator training feedback questionnaire**
To assess self-reported knowledge, understanding, skills and confidence to deliver the intervention and facilitator reflections on the training. Data were collated and averaged to provide an overall score out of 5 for each training block for knowledge and understanding (of the programme, facilitation techniques, developing a positive group dynamic and facilitator roles and responsibilities), skills and confidence (to present information, lead activities, create a positive group dynamic, manage time and difficult situations) (online supplemental document 3).

**Facilitator support calls**
To understand and respond to delivery challenges (two per site, one following week 1 and one halfway through the programme).

**Facilitator delivery checklists**
To understand the experiences and views on group delivery (weekly checklist, also included a check on content delivered) (module 1 example, online supplemental document 4).

**Focus group with facilitators**
To understand lead and assistant facilitator experiences and views on training, delivery and programme content (online supplemental document 5). All lead and assistant facilitators involved in delivering the programme were invited to attend the focus group, which took place at a meeting room within the University of Exeter. The 2-hour focus group took place once all groups were completed, led by JL and supported by BC-F (researchers not involved in delivering the programme). The focus group was audio recorded and transcribed verbatim (with any potentially identifiable information anonymised).

**Participants' feedback questionnaires**
To understand participant (control and intervention) experiences and views on programme content and delivery at the end of each group session (online supplemental document 6) and at the end of the programme (online supplemental document 7) and 6-month follow-up (online supplemental document 8).

**Participant phone interviews**
To understand and explore participant experiences, views and engagement with the group sessions and online materials (intervention and control) (online supplemental documents 9 and 10). Twelve intervention (two per group/site) and six control participants (one per site) were sampled to ensure that two out of the four male carers in the study (one control and one intervention) were interviewed and the range of parent carer challenges was represented (selection was based on lead facilitator comments and participant end-of-programme feedback). Interviews were carried out by a researcher not involved in delivering the programme (BC-F) and were audio recorded and transcribed verbatim, with names and other identifying information changed for confidentiality. Interviews took place as soon as possible after participants completed their postintervention measures and before the 6-month follow-up.

## Participants

Recruitment of participants and facilitators to the trial is reported above. For quantitative process evaluation data, all trial participants in both arms were sent online feedback forms after the programme, and all group participants were asked to complete the feedback forms after each session. For qualitative process evaluation data collection, participants were purposively sampled for maximum variation to include male and female perspectives, a range of engagement levels and parent carer challenges (see box 3), and all facilitators involved in delivering the programme were invited to a focus group.

## Data collection

Qualitative and quantitative data were collected from participants in both arms and facilitators on programme uptake, reach, training, delivery and experience as well as suggestions for improvement. Researchers (BC-F, JL) who were not involved in programme delivery collected the qualitative data. Box 3 presents the measures (details in online supplemental documents 3–10) and sampling process.

## Data analysis

### Qualitative data

All qualitative data were uploaded to NVivo V.12. Taking both an inductive and deductive thematic approach to analysis, we developed a coding framework to categorise the data from participant interviews, facilitator focus group, facilitator support calls and free text data from the participant and facilitator questionnaires. Three parent interviews were coded independently by three researchers (BC-F, AB, JL) initially and compared to agree the coding framework, which included combining codes and arranging them into higher level categories to organise the data. While the detailed codes were developed inductively, the categories were more deductive and followed the key areas of interest in the process evaluation, reflecting the interview topic guide. The agreed framework was then used to code the remaining interviews, adding new inductive codes when identified. Facilitator support calls and free text data from the participant and facilitator questionnaires were coded by AG and BC-F, and 20% were double coded by SM. All interview transcripts (n=18) were coded by BC-F, and 50% (n=9) were double coded by AB and JL. The focus group transcript was coded by BC-F and checked by JL and AB. The double coding and crosschecking of the coding, and regular team discussions on analysis and interpretation helped ensure the quality of analysis and minimise any potential researcher bias. Findings from the parent interviews were triangulated with data arising from the focus group and the questionnaires.

### Quantitative data

Relevant quantitative questionnaire data in relation to participant and facilitator experiences and views were collated and presented descriptively alongside the qualitative findings.

### *Data synthesis for intervention refinement*

Data relevant to programme improvement (ie, suggestions, critical comments, negative experiences) were extracted from NVivo. These data were coded and sorted into three categories: facilitator training, intervention delivery and intervention content (including online materials). For each suggestion/comment, the source of data was noted, including when there were mixed

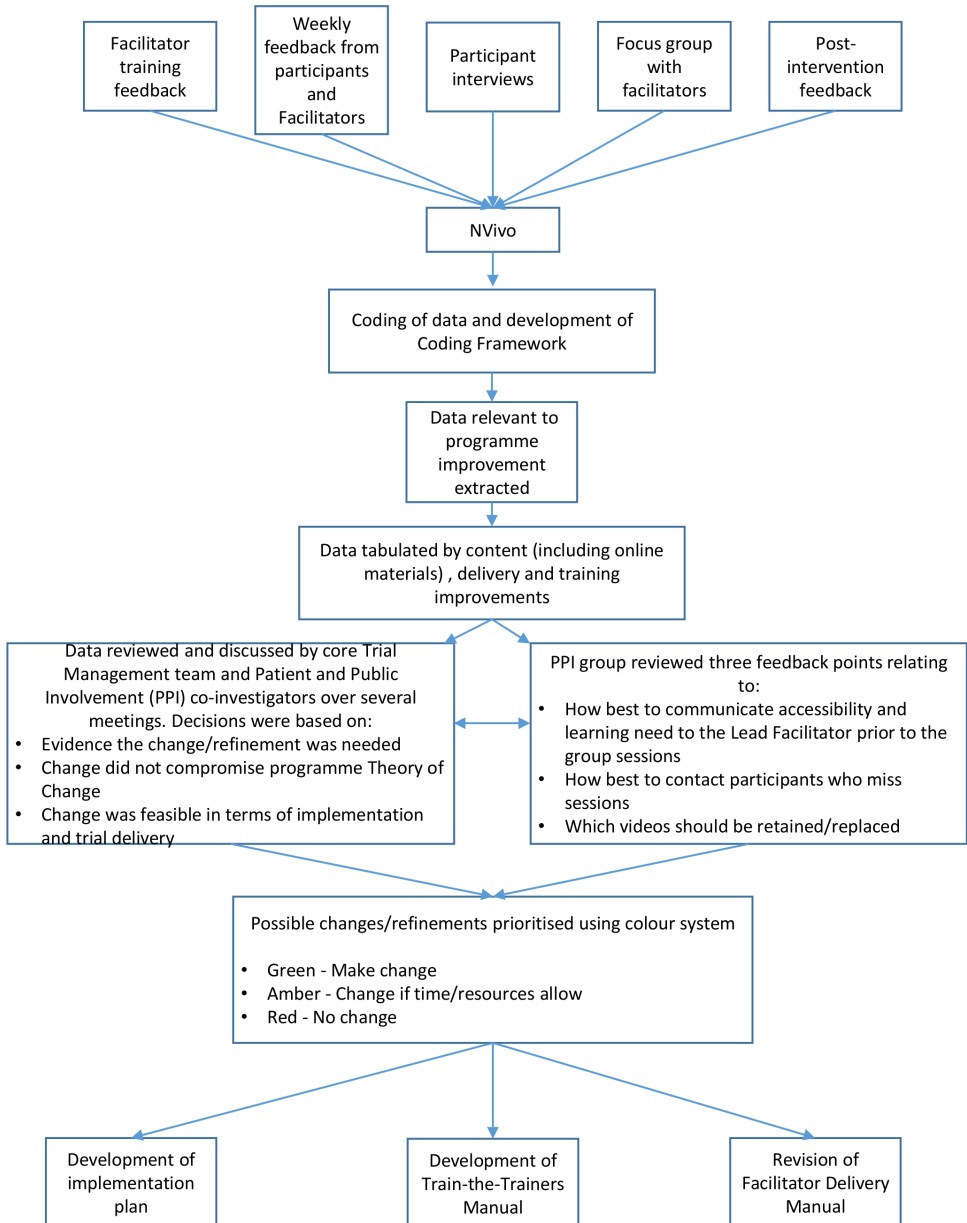

**Figure 2** Programme refinement process. LF, Lead Facilitator; PPI, patient and public involvement; TM team, Trial Management team.

or contradictory views. Figure 2 shows the refinement process. Potential refinements were discussed and prioritised using a colour-coded system.

## FINDINGS

Trial participants were aged 42.5 (8.0) years (mean (SD)), with 96% female and 97% white. Figure 1 shows the number of participants who provided each type of process evaluation data. Completeness of questionnaire data was high for participants (>80%) and facilitators (100%). According to facilitators' self-report delivery checklists, 90% of activities were delivered across all groups. Scores from researcher checklists of the nine audio-recorded modules[24] similarly indicated that 91% of activities were delivered. Interviews were carried out with

12 intervention (two per group/site; 11 female, 1 male) and six control participants (one per site; five female, one male). The focus group involved all lead (n=2 females) and assistant (n=6; 5 females, 1 male) facilitators. The mixed methods findings are summarised below, with illustrative quotes reported in boxes 4 and 5 and additional quotes and comments in online supplemental document 11.

### Facilitator experiences
#### Motivations and expectations
Five out of eight facilitators had experience of facilitating parent carer groups. Most reported being interested in delivering the programme because it fitted with their skills and interests, and offered an opportunity to further develop their professional skills and confidence.

## Box 4  Selected quotes illustrating facilitators' experiences

### Motivations and expectations

I feel that this training is a natural continuation to take forward the previous training I have been involved in. I am passionate about empowering parents with high quality information, support and advice. (Facilitator pretraining questionnaire)

For me it was a mixture of interlinked personal and professional reasons… I'm keen to do good in the world, if I can, keen to use some of my professional skills and enhance my professional skills while doing that. (Assistant Facilitator, focus group)

I know the power of groups and supporting each other… I know it's so important to have people that understand and support you, because it can be lonely. I had lots of hope that this would help other people and, yes, it definitely was fulfilled. (Assistant Facilitator, focus group)

### Training, preparation and support

Parent carers are a very specific group to deliver to and maybe something that is more specific around the baggage, if you'll excuse me for using that word. The baggage that parent carers bring into the room is extremely specific and the group dynamic isn't managing conflict, usually, with parent carers. It's very different. (…) I always find parent carers very quick to bond and support each other. Very, very quick; they really do move forward in that way. But when you've got an individual… Something about managing the individual rather than the group as a whole. (Lead Facilitator, focus group)

I didn't know enough about the programme before I actually joined in as an assistant… now, it would be fine going back but when you suddenly do the programme you haven't got a clue what's necessarily coming up next and you don't know how, emotionally, you're going to feel. (…) It probably would have been easier if I'd done the programme myself, because I could have then looked back at some notes, so any things that I come across being difficult, I would have had a clearer focus rather than what's actually going on with me at the time. (Assistant Facilitator, focus group)

Assistant Facilitator: For me, the support calls were more about the sharing of experiences—and the issues and part of the process and the research as well. I guess you [the researcher] were kind of in research mode to some extent in those calls. But the support seemed to me to be within the facilitator team.

Lead Facilitator: I agree absolutely with that. Totally. I think that the relationship that we had as co-facilitators is where I drew my support about anything that wasn't just on a practical basis.

Assistant Facilitator: And that's different levels of support needed. (Focus group)

### Programme content and group delivery

The repetitiveness of the way it was delivered really helped parents to predict and understand where they were going, so it put them in a very comfortable space as well. (…) Because every CLANGER is quite a repetitive process and it's unpicking the meanings, lots of self-reflection and we were using the collated feedback to constantly cross-reference and compare… (Lead Facilitator, focus group)

I found it hard work… and it was very intense and it was more counselling… because people were really disclosing lots of things about their lives and that way… the impact on us as facilitators was more intense experience… and whilst I think we didn't have anything we couldn't handle between us, it actually became more than the course is meant to be. (Lead Facilitator, focus group)

I think one of the things that I definitely felt was the relationship between the trainer and the assistant facilitator is amazing and I definitely

Continued

## Box 4  Continued

feel like the absolute bonus is I've gained a really good friend who I think going on it together actually benefits our health and wellbeing, just delivering it together. It is really positive. (Assistant Facilitator, focus group)

(See online supplemental document 11 for additional quotes/comments.)

Facilitators recognised the need for, and importance of, health and well-being training for parent carers, believed in the value of the programme (and group support) and wanted to support other parent carers. All reported interest in contributing to the research and programme development, and five hoped to benefit from the programme themselves.

### Training, preparation and support

Following blocks 1, 2 and 3 of training (box 2), all lead and assistant facilitators completed the facilitator training feedback questionnaire. Overall scores were out of 5. Knowledge/understanding and skills/confidence for each block were high at 4.5 and 4.6; 4.5 and 4.3; and 4.6 and 4.45, respectively. The only component to decrease over the training was the facilitator's skills and confidence to manage difficult/sensitive issues. Thus, unsurprisingly, in the focus group following programme delivery, they suggested the training could focus more on the delivery style and practice, and managing difficult issues specific to parent carers likely to arise in group discussions. Facilitators suggested that future facilitators could be recruited from those completing the programme, so facilitators would possess a greater understanding of the content and process and be able to build on their own experience of the programme. They also suggested better clarifying the roles and responsibilities of lead and assistant facilitators, and to be matched and trained together. They reported satisfactory support throughout programme delivery (ie, supervision from CDC, support calls with the research team), but particularly valued the sharing and support among fellow facilitators. Nearly three-quarters of facilitators believed that sufficient time had been allocated to training.

### Programme content and group delivery

Facilitators perceived the following benefits of the programme for participants: the focus on parent health/well-being (rather than their children); the face-to-face group delivery (rather than online); the consistent and predictable session structure; the weekly goal setting (but suggested simplifying 'SMART' (Specific, Measurable, Attainable, Relevant and Time based) to 'achievable' goals). They perceived both 6-week and 12-week delivery as suitable.

Facilitators found the practical activities an important component (in addition to group discussions) but some activities triggered difficult feelings among some participants. The facilitators reported that some participants disclosed difficult personal issues, which were challenging

**Box 5    Selected quotes illustrating participants' experiences**

**Motivation to participate and expectations**

I feel that being a parent-carer is really draining on your mental health and also I wanted to try and get healthy anyway, especially with it being January. (…) a bit of knowledge maybe about the way I feel and maybe a bit less guilt, the fact that I feel that I need to take time out for myself. (P13, control)

I just thought it could be useful because a lot of the time I find as a parent carer all the focus is on the children which is obviously the main thing but the parents need to be healthy and happy to give the children the best. (P8, intervention)

**Experiences of the programme and its impact**

It was fantastic. I think it was really well done. (…) I think it's changed my life. (…) I think the best thing was going to the group, meeting everybody in the group, just meeting like-minded people, having a break from life and being in a different place for a while, reflecting on aspects of my life that I didn't really think about or had put away, and I learnt more about myself. (P5, intervention)

Meeting people who understand and 'get it' reduced my isolation. Having parent carers as facilitators helped a lot. Having 12 structured sessions made me commit to it and focus on my own health. Having structured fun informative sessions helped a lot—I definitely went through a process of change. The use of humour was really important to me—and learning to be more resilient. (End-of-programme feedback form)

A lot of it is very common sense stuff. (…) It's like a revelation without it being a revelation, because you know all that stuff, but you don't take the time to think about it and it was just very much about focusing on us and improving things for ourselves before we can do it for everyone else really. (P3, intervention)

Unfortunately, I think we were one of the groups that was very small, which was good in some ways because you got to know the people better and you had a bit more time, but then on the other side, at one point there was only two of us there and you've not got everybody's stories, you've not got everybody's experiences, it was just a couple of us. (P4, intervention)

If the group was a little bit bigger…you would get more input and there would be more discussion because even though we were able to discuss things, I think a bit more of an open, broader amount of different people with different lives would make it a lot more interesting. (P6, intervention)

Was there anything that I didn't like? I think the only thing that made me feel… was sometimes like the icebreaker thing, but that's just me feeling nervous about (…) sometimes in group situations if I'm asked my opinion on something or asked to think about something and you've got to kind of think quickly, I get really nervous. (…) I think at the beginning also you are worried about what will people think of me and how will I be perceived by other people. You don't want to say the wrong thing… (P2, intervention)

**Views on online materials**

Initially I was curious and then I think it dwindled off because I felt like the course was covering everything so thoroughly that would it add anything to what I had already had? (P4, intervention)

What was there was absolutely fantastic and I think the videos were a good way of doing it because I am better at learning through watching the videos and being able to go back to them rather than just having loads and loads of information to read… (P18, control)

Continued

**Box 5    Continued**

I found doing the online course quite isolating. Personally, I would have made more changes if I had been in the group. (…) I have ticked the box, I have done it (…) I found myself thinking about everything else that I should be doing, whereas if I had gone to the group and they were watching that video… you are switched on to be watching that video, whereas for me, I was there thinking, 'Okay, I have got to do this, so I will sit down and do this'… and then my mind… at one point, I was so tired that I was thinking, 'Oh gosh, I have got to go and make lunch'… (P18, control)

**Factors influencing ability to engage with and benefit from the programme**

…on some days you are impacted by inadvertent events and there is no relaxation, there is five hours' sleep. That's just the way it is. (P14, control)

I have some really quite big social care needs and health care needs of my own that are not being met and so it is much more difficult to sit there going, 'Ah well, have I eaten properly today?' But those things don't exist so it's better than nothing. (…) But I think it's working really well for some people. Yes, I can see that other members of the group were getting masses more out of it than I was. So I think if it's the right thing for you then it's clearly way more beneficial. (P2, intervention)

I couldn't really make out whether I liked it or not. It was alright, it was pleasant enough, but I think I've got a whole bunch of underlying issues that are not being addressed… for some people it was clearly really hitting the spot… and for me, I'm going, if I don't actually get some proper mental health care and social care support I can't really look… well I can look at it but I can just go, 'Yes, that will be nice, one day, maybe, mmm, okay'. (…) (Interviewer: What did you not like or not find helpful, if anything?) I suppose the assumption that's it in my power to change some of these things when it just isn't. (P2, intervention)

to address in the group without turning into counselling sessions; thus, the facilitators stressed the importance of training in managing such situations. This was more problematic when groups were small, and the facilitators agreed that six to eight participants were an optimal group size. The facilitators emphasised the importance of facilitators being parent carers, and how the positive experiences of cofacilitating and the programme helped them reflect on their own health and well-being.

Facilitators thought that each session went well, and that participant engagement was high, their average ratings across groups for each session ranging from 4.4 to 4.8 out of 5 for how well they thought the session went and 4.5–5 out of 5 for parent engagement. Facilitator judgement of how connected their groups were following each session was very high and increased over the course of the programme from 5/7 to 7/7. Box 4 presents the selected quotes illustrating facilitators' experiences.

**Participants' experiences**

Below we refer to 'interviewees' to indicate the views only of those interviewed and to 'participants' to indicate data collected from the questionnaires. Box 5 presents the selected quotes illustrating participants' experiences.

### Motivation to participate and expectations

All interviewees (n=18) reported participating in the programme to focus on and find ways to improve their health and well-being. Some wanted to do that to better support their families, and to feel less guilt for taking time to take care of themselves. A few interviewees thought the programme took place at the right time for them: when they were able, ready and available to participate and focus on their health. Most interviewees also reported wanting to contribute to research and help other parent carers, especially as they perceived little available support for parent carers.

The majority of interviewees in the group programme reported positive aspects of face-to-face delivery, such as peer support and dedicated time to share and discuss issues affecting their well-being. A few participants, mainly from the online-only control arm, reported that the flexibility of the online programme enabled them to engage with the programme more easily than attending face-to-face groups. Overall, group-based delivery was preferred, including those who had not experienced it, as groups were perceived to instigate a stronger commitment to engage with the programme.

### Experiences of the group programme and its impact
#### Positive experiences

Overall, 84% (37/44) of participants were satisfied or very satisfied with the group programme, with 67% (28/42) finding it useful in helping them to improve their health and well-being. Most of the interviewees echoed this, noting the positive impact of the programme on them and their families. Two reported that it 'pushed them out of their comfort zones' by addressing some uncomfortable but important issues. Several participants commented that the programme should be rolled out to benefit all parent carers.

#### *Groups*

All participants who responded to the postintervention questionnaire reported feeling included and part or very much 'included and part of the group' with 85% (17/20) indicating that session length was 'about right'. All intervention interviewees described one of the main benefits of the programme as having the opportunity to discuss and share with other parent carers in a supportive, empathetic, safe and respectful group context. They highlighted the value of peer support and discussions in motivating change. Interviewees also perceived the group programme as providing more than support groups because of the practical activities and exploring barriers and solutions in specific areas. Some were surprised by the small group size but liked it as it enabled them to participate and get to know each other and bond as a group; yet, most thought that slightly larger groups would provide more varied perspectives. Interviewees noted also that the group programme gave them time to focus on themselves and permission (to take time) to take care of themselves.

#### *Peer facilitators*

All participants who responded to the questionnaire were satisfied or very satisfied with the way the programme was delivered, and interviewees highlighted the critical importance of the facilitators being parent carers, enabling shared understanding and empathy. Participants valued facilitators delivering the programme together ('complementing each other' and 'bouncing off each other'), facilitating learning through group discussions (rather than 'teaching'), creating positive group context (with facilitators seen as part of the groups) and being knowledgeable, understanding and kind.

#### *Content and activities*

Interviewees generally had positive comments about programme content and preferences for different activities. The content and activities were mainly valued for providing structure and prompts to group discussions and focus on different life areas. The promoted messages were seen as 'common sense' and applicable to all people, but more challenging to parent carers. The programme reminded participants about the importance of health/well-being, and discussing/sharing reinforced that, thus prompting them to make changes.

#### *Reflecting and setting small achievable goals*

Interviewees valued the programme's focus on simple and meaningful actions (CLANGERS) that increased their awareness of areas of their lives in which they could make positive changes (while also helping accept things that they could not change). Interviewees also valued focusing on small steps that they can take, setting achievable goals at each session/module and then reflecting on them. This helped them feel more in control of doing something positive about their health/well-being. Seventy-six per cent (35/46) reported making changes.

#### *Less positive experiences*

Despite overwhelmingly positive experiences, nine participants did not find the programme useful, with five indicating ambivalence. A few interviewees reported a less positive experience and *not* making changes, mainly due to factors outside the programme (described below); however, they still valued the raised awareness and 'hope' it provided. This was reflected in the quantitative feedback with 24% (11/46) reporting not making changes. A few interviewees reported feeling apprehensive and nervous at the start of groups; finding different activities or content difficult, challenging or less appealing; feeling uncomfortable with others in the group (eg, due to expressed views/comments); and perceiving less shared experience and challenges with others in the group (eg, due to personal or system-related contexts).

#### Views on online materials

Participants' experiences of, and views on, using online materials were mixed. Interviewees attending the groups reported finding the online materials unnecessary or just reinforcing the sessions. Interviewees from the control

group reported some positive experiences of the online-only programme (valuing videos as an engaging way of providing information) but found some content less relevant or helpful. Overall satisfaction of control participants with the programme was good, although lower than intervention participants (66% vs 84%), with only 38% (11/29) finding the programme 'useful in helping them to improve their health and wellbeing', although 65% (20/31) reported making changes.

Interviewees in both study arms described the importance of group discussions in enhancing the learning and programme impact. Without group sessions, some found the online-only programme isolating (without opportunities to discuss, share ideas and problem solve with other parent carers) and harder to focus on because of lacking reminders and scheduled time (with other matters taking priority).

### Factors influencing ability to engage with and benefit from the programme

Interviewees in both study arms reported factors that affected their ability to attend the sessions or use online materials, including: other (unexpected) commitments, lack of time, inadvertent events (their or children's illness), access/transport and childcare. A few also discussed external factors out of their control (eg, social care needs, access to respite, work situation) that affected their well-being and ability to benefit from the programme, reinforcing the importance of participating in the programme at the right time in their lives.

### Programme refinement

The process evaluation confirmed that the HPC programme was highly acceptable to both parent carers and facilitators with the vast majority expressing positive experiences; however, less positive experiences and suggestions were carefully considered and used in the refinement process (figure 2). Several potential areas for improvement were identified and, where possible, incorporated into the refined HPC programme. Key changes are outlined below.

#### Optimising session and online content
► Wording of the online materials simplified to increase understanding, engagement and usage.
► Following suggestions and less positive views on goal setting, 'SMART' replaced with 'achievable' goals, and examples of SMART goals were added to online materials and delivery manual to increase understanding.
► Create or select videos more relevant to parent carers (as some participants were less positive about some of the more generic health-related videos).

#### Optimising training
► Explanation of the rationale for certain activities included to increase understanding of their purpose.
► Training in managing challenges that specific content/activities may generate, and worked examples on how these may be addressed.

► Training in how best to present and deliver the videos to elicit discussion around the key 'take away' messages.
► Facilitators to complete the online modules prior to training so that they are familiar with the CLANGERS and resources, providing more time to focus on delivery strategies.
► Group dynamics session refined so that it is more practical and interactive (scenarios and practice included in how to build, enhance and maintain group cohesion under challenging situations; how to find commonalities in shared experience despite having differing challenges/situations).
► Delivery process for each of the CLANGERS to be modelled in detail using 'Connect' (as it is the most challenging of the CLANGERS to deliver and experience) to ensure that facilitators understand the application of the programme's theory of change.
► Trainers asked to create and disseminate a 'Frequently Asked Questions' document to follow-up any questions/concerns not addressed in the training due to time constraints.
► Increased focus on developing delivery skills using modelling techniques to improve confidence and quality of delivery (eg, modelling good responses to parent questions, supporting parents who are struggling).
► Roles and responsibilities of the lead and assistant facilitators clearly explained to avoid misunderstanding and enhance team working.

#### Optimising delivery
► Lead and assistant facilitators to communicate with each other prior to the first session (using the 'preparation' checklist) to allay any delivery fears/concerns.
► Prior to session 1, to allay any participant concerns before joining the group, facilitator photo, with a written introduction to be sent to participants; lead facilitators to call participants to introduce themselves, ascertain any practical support required and take steps to provide this support where possible.
► Advice on managing difficult situations added to the delivery manual.
► Description of roles and responsibilities of the lead and assistant facilitators added to the delivery manual.
► Key 'take away' messages from each video added to the delivery manual.
► Delivery manual adapted to accommodate the 12×2 hour format to improve coherence.
► Completion of a simple participant feedback form at the end to provide information to improve future group delivery.
► Following participants' and facilitators' less positive comments about smaller groups, change minimum viable group number from 4 to 6 to increase the range of perspectives and create better conditions for peer-to-peer learning and support.

## DISCUSSION

This paper presents facilitator and parent carers' views on and experiences of the HPC programme from the feasibility RCT, which have informed the refinement of the programme. We are now exploring implementation uncertainties, funded by the Economic and Social Research Council Impact Accelerator Account Strategic Initiative Award (ES/T501906/1) in preparation for a further evaluation of programme effectiveness. Fidelity to delivery and qualitative data show that the training of facilitators was successful. The group programme was valued for providing peer support and practical activities, where difficult and emotional conversations are facilitated and explored. This is a key function of the programme, which accounted for the high levels of satisfaction and reported impact on both the parent and the wider family. It appears that this aspect of the programme necessitates larger groups (6–12) to allow for varied perspectives and a 'facilitation' rather than 'counselling' approach. Group size, therefore, will be explored as part of intervention fidelity to function[25 26] in the definitive trial.

As intended, consistent with our logic model, participants reported that the programme increased awareness of what parent carers could and could not change and their self-efficacy to engage in health-promoting behaviours (CLANGERS). Participant motivations and expectations showed that, overall, the target group was reached (ie, those who wanted to and reported feeling ready to do something to improve their own health/wellbeing), with the intended mechanisms of action (social identification and peer support) matching the participants' expectations and experiences.

Facilitators reported that the relationship between the leads and assistants was important for effective delivery and that clarity on and practice of these different roles were a crucial part of training and quality delivery. Both intervention participants and facilitators thought that, to deliver the programme effectively, leads and assistants needed to have completed the programme themselves and that the development of a network of facilitators to share experiences of delivery using support calls and/or online meetings was important in supporting them to improve their practice.

The strength of this research is that it has systematically followed key principles of intervention development[27 28] and refinement using a dynamic, iterative and creative process with extensive stakeholder consultation, where the developers have been open to change based on data collected in a series of iterations.[22 23] We examined how the intervention will be evaluated in the next phase of research and identified learning and key uncertainties to be addressed, such as blended (online and face-to-face) delivery of group sessions, commissioning and implementation. Reporting mixed methods data on participant views and experiences and the subsequent refinement process in feasibility studies is recommended.[29] It increases knowledge about the intervention refinement/ optimisation process and allows linkage of intervention development processes and subsequent trial outcomes.

However, the study has limitations. The lack of ethnic diversity in South West England, where this study was conducted, coupled with the low representation of men as 'primary care givers' (an inclusion criterion) meant that the sample was gender and ethnically homogeneous. Experiences and views, therefore, may not represent fathers or parent carers from different cultures and contexts.

Most work on culturally and linguistically diverse groups and parenting interventions has been conducted in the USA and is equivocal about whether outcomes differ by ethnicity.[30 31] However, adapting interventions for different ethnic groups poses many issues. Adaptation may neither be practical nor a desirable service model for multiethnic European cities.[32] Presently, we do not have the available data as individual trials are not powered to test intervention effects by ethnicity.

Nevertheless, we will explore in future evaluative work with mixed and single ethnic groups how ethnicity, social disadvantage, gender and/or other personal factors might intersect to exacerbate the health issues arising from being a parent carer,[33] and affect the development of a shared group identity and implementation more generally.

This study has helped to refine the programme in many aspects; however, there remain a number of barriers to parent carers' capacity to engage in all group sessions and benefit from the programme. We need to explore these barriers in further detail and the extent to which they can be mitigated to enhance accessibility to participate in the programme. As we write, the COVID-19 pandemic has meant social distancing and increased challenges to running groups. We are therefore keen to explore to what extent the group programme could be delivered virtually using videoconferencing, which may overcome some other barriers to participation, but we would need to evaluate the extent to which peer support and cohesive support of the group are maintained.

## CONCLUSION

The format, content and delivery of the HPC programme was highly acceptable to participants and for facilitators to deliver. The process evaluation data enabled programme refinement to optimise impact going forward. Although the programme focuses on promoting health and wellbeing at an individual level (ie, individual psychological and behavioural change), we acknowledge the importance of other factors at interpersonal, community and societal levels that affect parent carers' health and wellbeing, such as access to services, negative public attitudes towards disability, which in turn impact on parent carers' capacity to make and sustain changes. The programme does, however, provide support and hope for those who find it difficult, both practically and psychologically, to attend to their own well-being. The challenge we face

going forward is delivery of the programme in the shadow of COVID-19. The team plan to explore how this might be reimagined to accommodate a new way of supporting parents, minimising risk to health, while delivering an accessible and inclusive package that maintains fidelity to function.

**Author affiliations**
¹NIHR Applied Research Collaboration South West Peninsula (PenARC), University of Exeter Medical School, University of Exeter, Exeter, UK
²Relational Health Group, Institute of Health Research, University of Exeter Medical School, University of Exeter, Exeter, UK
³Peninsula Childhood Disability Research Unit (PenCRU), University of Exeter Medical School, University of Exeter, Exeter, UK
⁴Medical Sciences Division, Nuffield Department of Primary Care Health Sciences, Oxford University, Oxford, UK

**Acknowledgements** We thank the members of the PenCRU Family Faculty and the Stakeholder Advisory Group for their involvement in the development of the intervention and the design and execution of this study, and the Council for Disabled Children for their collaboration. The authors are grateful to all participating parent carers and their families and the facilitators who delivered the programme. We thank our colleagues within the wider PenCRU team: Katharine Fitzpatrick for organising the parent carer working group meetings and Tania Hind for administrative support. We also acknowledge the support from the National Institute for Health Research Clinical Research Network (NIHR CRN) and the National Institute for Health Research Applied Research Collaboration South West Peninsula.

**Contributors** CM led the development and preliminary evaluation of the programme and was the principal investigator of this study. GB managed the project, including overseeing day-to-day recruitment and data collection. GB and CM drafted the initial study design with input from AB, JL, VB, MT, MF, AM and SL. AB, AM, MF and CM designed the original programme. KW recruited the programme facilitators and arranged the delivery sites. GB, BC-F, KW and AG recruited the participants and facilitated the data collection. AM, MF, BC-F and KW planned, prepared and delivered the facilitator training and support. JL led the programme refinement process supported by GB, AB, VB and MT. AB, BC-F, AG, SM and JL coded and analysed the qualitative data. SL facilitated the infrastructure support through the National Institute for Health Research Applied Research Collaboration South West Peninsula. All authors served on the Trial Management Group, contributed to drafting this paper and approved the final manuscript. The study sponsor is the University of Exeter.

**Funding** This paper presents an independent research funded by the National Institute for Health Research (NIHR) under its Research for Patient Benefit (RfPB) programme (grant reference number PB-PG-0317-20044). Delivery of the Healthy Parent Carers programme was funded by a grant from the National Lottery Community Fund through the Reaching Communities programme (grant number 10343962) and supported by the Public Health England, the Kernow Clinical Commissioning Group and the Northern, Eastern and Western Devon Clinical Commissioning Group. Peer review of the study design was provided during the NIHR funding application process.

**Disclaimer** The funders have had no involvement in the writing of the manuscript and will not have any role during its execution, analyses and interpretation of data, writing of the report or the decision to submit the report for publication. The views and opinions expressed in this paper are those of the authors and not necessarily those of the National Institute for Health Research or the Department of Health and Social Care.

**Competing interests** None declared.

**Patient consent for publication** Not required.

**Ethics approval** The study procedures were approved by the University of Exeter Medical School Research Ethics Committee in the College of Medicine and Health (UEMS REC 18/06/174). All participants in the study provided written consent to participate.

**Provenance and peer review** Not commissioned; externally peer reviewed.

**Data availability statement** Data are available upon reasonable request. The datasets analysed during the current study are available from the corresponding author on reasonable request.

**ORCID iDs**
Jenny Lloyd http://orcid.org/0000-0003-3211-5972
Aleksandra Borek http://orcid.org/0000-0001-6029-5291
Christopher Morris http://orcid.org/0000-0002-9916-507X

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
