## [Reviewer comments · BMJ Open]

ARTICLE DETAILS

TITLE (PROVISIONAL)	The Healthy Parent Carers Programme: mixed-methods process evaluation and refinement of a health promotion intervention
AUTHORS	Lloyd, Jenny; Bjornstad, Gretchen; Borek, Aleksandra; Cuffe-Fuller, Beth; Fredlund, Mary; McDonald, Annabel; Tarrant, Mark; Berry, Vashti; Wilkinson, Kath; Mitchell, Siobhan; Gillett, Annette; Logan, Stuart; Morris, Christopher

VERSION 1 – REVIEW

REVIEWER	Jessica Ward Children's Hospital of Los Angeles
REVIEW RETURNED	03-Jan-2021

GENERAL COMMENTS	Healthy Parent Carers feasibility randomized controlled trial: process evaluation and refinement of a health promotion intervention for parent carers of disabled children. Title: The title is clear and conveys the main intent of the manuscript. Overall: This work is unique, addresses a gap in the literatures and is an important contribution to the field. This manuscript is well-written, although there are opportunities to improve the readability. The figures and tables are clear and generally convey the feasibility findings. The manuscript would be strengthened with the addition of preliminary efficacy data. The supplementary paper published in Pilot and Feasibility Studies includes multiple outcome measures that were administered to participants at 3 time points (pre-randomization, immediately post-intervention and 6-months post). This data should be available on the majority of participants randomized. Is this data going to be presented separately? If so, that should be explicated in the Discussion section. Figure 1: Missing information at T3 (number of participants evaluated). Methods: page 8, line 46: seem to be missing a word. "These data were (_) and sorted into three categories.... Results: Consider presenting the quantitative findings in a single table to clearly reflect the parent feedback (both weekly and overall) and follow-up questionnaires. It would be instructive to report the intervention fidelity data generated from the facilitator checklists. Did the sessions adhere to the prescribed content? If so, to what extent? Discussion: Provide additional context for the summary of findings. How do these findings compare with relevant, existing research? Consider including a section on Future Research.
--

REVIEWER	Helen Bourke-Taylor Monash University Faculty of Medicine Nursing and Health Sciences, School of Primary and Allied Health Care
REVIEW RETURNED	18-Feb-2021

GENERAL COMMENTS

Dear authors,

Well done on a great manuscript and moving this important field of research forward. Ultimately the paper will join the growing evidence that aims to change practice and make health and wellbeing programs for carers standard practice. Please make adjustments to the structure of the paper so that it is more aligned with conventional reporting.

At present the manuscript is more like a report that is easy to read and understand as someone who is very aware of the purpose, needs and content of a program for parent carers, but the novice reader would benefit from clearer convention headings like intro ending in research questions/aims, methods, (usual sections), results etc results etc.

This paper should be prioritised for publication in a medical journal to advocate for carers who provide the vast majority of complex care in families that are also serviced by medical/allied health and other disability services. The paper is not directly related to the children with disabilities/chronic medical conditions and therefore this article is at risk of rejection. However the health and wellbeing disparity experienced by carers is large and confirmed in numerous international studies. If carers are not well, all evidence based medical/allied health and available services cannot be implemented and therefore the downstream effects on the care recipient is mammoth. Hence I strongly recommend publication of this article in this journal. It is relevant. It is timely in the era of COVID-19 where families carers face even more pressure. Having made this point, the authors need to make a stronger argument about the crucial nature of this sort of program to practitioners, service providers and policy makers. Also consider findings from Bourke-Taylor, H.M., Lee, D.C.A., Tirlea, L., Joyce, K., Haines, T. Support for the mental health of mothers of children with a disability: Systematic review, meta-analysis and description of interventions. Journal of Autism and Developmental Disorders (early view) doi: <http://10.1007/s10803-020-04826-4>.

I have made minor suggestions in comment boxes in the attached PDF.

Please consider the following:

Abstract, make clearer the data collection points--pre and post intervention or just after participation?

Include gender ratio in abstract and paper in general. Is this paper mainly about mothers? Please explain.

The paper needs better alignment between the title (RCT-- Feasibility) and the design described. At times this reads like a pilot study, except for the mixed methods and other components. Clarity and consistency is needed to define the methods and type of study and the actual study. There does not appear to be a flaw in design-- just better explanation and consistency needed from title to abstract to manuscript content. i.e. a "A pre-planned mixed methods process evaluation" is not the same as a feasibility study.

Consider use of terms used in feasibility study--acceptability, usability, content relevance, trial of outcome measures suitable for the RCT....

Minor word changes suggested to increase scientific reporting--i.e. word 'felt' replaced with 'reported'; page 6 "they often" might be replaced with "irrefutable evidence indicates that..." and cite sources, also qualitative results remove word 'they' and refer to participants or facilitators when describing who said what.

The program itself is potentially effective at directing and empowering parent carers to change their own behaviour to improve their health outcomes. The targets for the program seem to be healthy behaviour change impacting health. This needs to be written more directly. The program itself can't improve health. The program promotes behaviour change to help participants achieve better health themselves.

The TiDIER checklist is a very useful checklist to describe all of the aspects of an intervention. This might assist authors to create a summary of all of the parameters of the intervention--replacing Box 1. The supplementary table doc 1 is detailed and excellent but wondering if you need a better summary in the published article because many readers won't download this?

In key points--define your homogenous population and state that application to CALD populations requires future consideration and participatory involvement.

In the description of intervention mapping the process is described as if it was iterative and inductive BUT it also seems that the CLANGERS program was applied over the top of this. Consider whether you were guided by CLANGERS and feedback in the way this is described for consistency.

If the facilitators were trained--please explain how. What risk management protocols were put in place and how were they included--i.e. carers with mental health issues managed within the group, directing parents to medical assistance, domestic violence issues...other? If this doesn't exist, describe this under future developments because the evidence is that domestic violence is higher and children with disability have a 4 fold risk for abuse and neglect related to parental mental health--another alarming statistic that might be included to justify the crucial nature and need for your program.

We cannot assume that peer mentors or leaders know what they are doing just because they have lived experience--this aspect needs explaining as the effectiveness is surely related to the skills of the facilitators. i.e. effective groups have been run by health professionals (see systematic review above) but little evidence that peer mentors are as effective in behaviour change. Peer mentors and leaders are effective for support but this intervention aims to change healthy behaviours. Please explain or add more to box 2 (i.e. safeguarding and facilitators need more explanation and also who trained facilitators and why they were sufficiently skilled to do so).

Please add in research aims or questions clearly with a heading to draw the readers attention

Methods that you implemented to support the trustworthiness of the qualitative data is needed to show dependability of your qualitative findings. The authors present the summary of what participants said in their own words and the reader doesn't know how trustworthy this is--i.e. what evidence can you provide which will show that

	researcher bias what minimised in the analysis of interviews. This is important to explain. The qual results appeared to be handled deductively with specific feedback sought. Please include a description of the qualitative design and cite the source that drove the qual data analysis process. Reformat the methods and results sections so that they are sequential and not blended as they currently are. This is confusing for the reader and detracts from the rigour that you applied to the research. For example, under qual study--describe methods and then results for who and how many applied etc. Who interview participants and how did you control for positive reporting if interviews were not anonymised. This applies to the quant data too. Were participants able to be anonymous and honest or is there a risk that participants were positively reporting. Add in all ethical approvals and considerations--the study is about a vulnerable group but ethics is not explained--please add this to the manuscript. The research team are to be congratulated on an interesting and thorough study that has obviously included co-design and co-delivery with the target group. The 14 participants reporting neutral or not satisfied with the group were potentially a rich source of feedback to make changes to improve the program. Please expand what was learned form these questionnaires or interviews. What did participants think of the instruments and data collection etc --i.e. feasibility of research design as well as the content and delivery of the program. This seemed to be the intention of the study but results/ explanation may need to be added. The impact on this small group also relates to safeguarding and the skill of facilitators. Please show how harm can be minimised in future programs form these learnings. You say "however less positive experiences were carefully considered in the refinement process" but please explain how. Please add some negative but constructive quote from participants into box 5 to represent these experiences (if possible) Discussion is excellent and brings paper together for the reader. Supp doc 11 is extensive but worth including. More cross referencing to all supp docs, tables needed. I am looking forward to seeing this paper in print and also final results when they are ready. Thanks for the opportunity to read the manuscript. The reviewer provided a marked copy with additional comments. Please contact the publisher for full details.
--	---

VERSION 1 – AUTHOR RESPONSE

Reviewer 1

Reviewer: 1 - Dr. Jessica Ward, Children's Hospital of Los Angeles Comments to the Author:

Title: The title is clear and conveys the main intent of the manuscript.

Overall: This work is unique, addresses a gap in the literatures and is an important contribution to the field. This manuscript is well-written, although there are opportunities to improve the readability. The figures and tables are clear and generally convey the feasibility findings.

The manuscript would be strengthened with the addition of preliminary efficacy data. The supplementary paper published in Pilot and Feasibility Studies includes multiple outcome measures that were administered to participants at 3 time points (pre-randomization, immediately post-intervention and 6-months post). This data should be available on the majority of participants randomized. Is this data going to be presented separately? If so, that should be explicated in the Discussion section.

Thank you for raising this and sorry that it was not clear that these data will be presented separately. This partner paper has now been submitted and we have added the reference where there is a statement at the end of the introduction (Page 6, line 3) reporting that 'trial findings related to recruitment and retention, fidelity of intervention delivery, the feasibility and acceptability of trial processes and outcome data will be reported separately'. We think that this provides sufficient clarity.

Figure 1: Missing information at T3 (number of participants evaluated).

This information has been added to the T3 box, Figure 1.

Methods: page 6, line 46: seem to be missing a word. "These data were (_) and sorted into three categories...."

Missing word 'coded' has been added, page 7, line 22

Results: Consider presenting the quantitative findings in a single table to clearly reflect the parent feedback (both weekly and overall) and follow-up questionnaires.

This was considered by the authors, but we decided that integrating these data with the qualitative findings enabled us to present the whole rather than the sum of the individual qualitative and quantitative parts for each section, an intentional process when using mixed methods in health research. As a result, we believe that reader understanding is improved for having a strong and coherent narrative.

It would be instructive to report the intervention fidelity data generated from the facilitator checklists. Did the sessions adhere to the prescribed content? If so, to what extent?

This information has been added to the findings (Page 7, line 31)

Discussion: Provide additional context for the summary of findings. How do these findings compare with relevant, existing research? Consider including a section on Future Research.

We have now made reference to two recent systematic reviews in the introduction rather than the discussion as the focus of the study was to present how we used the mixed methods findings to refine the programme content, training and delivery rather than compare our findings (i.e. experiences of parent/facilitators of the HPC) to other parent carer interventions. (Page 4, line 17).

Future research considerations have been added to the discussion (page 14, line 28).

Reviewer 2 - Dr. Helen Bourke-Taylor, Monash University Faculty of Medicine Nursing and Health Sciences Comments to the Author:

Well done on a great manuscript and moving this important field of research forward. Ultimately the paper will join the growing evidence that aims to change practice and make health and wellbeing programs for carers standard practice.

Thank you

Please make adjustments to the structure of the paper so that it is more aligned with conventional reporting. At present the manuscript is more like a report that is easy to read and understand as someone who is very aware of the purpose, needs and content of a program for parent carers, but the novice reader would benefit from clearer convention headings like introduction in research questions/aims, methods, (usual sections), results etc results etc.

We advocate that the present structure is appropriate for the aims of this study and provides a complete and transparent report. Furthermore, this paper is complying with ICMJE reporting guidelines <http://www.icmje.org/recommendations/browse/manuscript-preparation/preparing-for-submission.html> (abstract, introduction, methods, results/findings, discussion and references). We have, however, changed the heading 'background' to 'introduction' (page 4, line 10) and in the methods and findings sections have added information on the participants (Page 6, line 16 and page 7, line 29 respectively).

This paper should be prioritised for publication in a medical journal to advocate for carers who provide the vast majority of complex care in families that are also serviced by medical/allied health and other disability services. The paper is not directly related to the children with disabilities/chronic medical conditions and therefore this article is at risk of rejection. However the health and wellbeing disparity experienced by carers is large and confirmed in numerous international studies. If carers are not well, all evidence based medical/allied health and available services cannot be implemented and therefore the downstream effects on the care recipient is mammoth. Hence I strongly recommend publication of this article in this journal. It is relevant. It is timely in the era of COVID-19 where families' carers face even more pressure. Having made this point, the authors need to make a stronger argument about the crucial nature of this sort of program to practitioners, service providers and policy makers. Also consider findings from Bourke-Taylor, H.M., Lee, D.C.A., Tirlea, L., Joyce, K., Haines, T. Support for the mental health of mothers of children with a disability: Systematic review, meta-analysis and description of interventions. Journal of Autism and Developmental Disorders (early view)

<https://link.springer.com/article/10.1007/s10803-020-04826-4>

Thank you, we agree that we need to make a stronger case for the intervention. In the introduction we have attempted to do this within the constraints of the overall word limit. (Page 4, line 12).

I have made minor suggestions in comment boxes in the attached PDF.

We have not been able to access the pdf.

Please consider the following:

Abstract, make clearer the data collection points--pre and post intervention or just after participation?

Data collection points have been added to the abstract. (Page 3, line 13).

Include gender ratio in abstract and paper in general. Is this paper mainly about mothers? Please explain.

Gender percentages, including ethnicity, have been added to the abstract and the findings (page 3, line 11 and page 7, line 29 respectively) and figure 1. In the findings (page 8, line 3) section, we have added in the gender split for the interviews and for the focus group.

It was not unexpected that participants were mainly mothers. Men are less represented as parent carers. We do not exclude males, rather the sample bias towards women reflects that mothers are primary carer givers (primary carer was an inclusion criterion), thus both parents were excluded from participating as it was believed this would affect the group dynamic. Future research will explore how best to encourage fathers to participate.

The paper needs better alignment between the title (RCT--Feasibility) and the design described. At times this reads like a pilot study, except for the mixed methods and other components. Clarity and consistency is needed to define the methods and type of study and the actual study. There does not appear to be a flaw in design--just better explanation and consistency needed from title to abstract to manuscript content. i.e. a "A pre-planned mixed methods process evaluation" is not the same as a feasibility study.

We have adjusted the title accordingly to better reflect the focus of the paper.

Consider use of terms used in feasibility study--acceptability, usability, content relevance, trial of outcome measures suitable for the RCT....

This paper is presenting mixed methods process data on the motivations of parent carer participants and facilitators to take part in the Healthy Parent Carer feasibility trial, their views and experiences of receiving/delivering the programme and its subsequent refinement in response to these data (end of the introduction, page 6, line 4). These terms relate to our recently submitted partner paper rather than this paper.

Minor word changes suggested to increase scientific reporting--i.e. word 'felt' replaced with 'reported'; page 6 "they often" might be replaced with "irrefutable evidence indicates that...." and cite sources, also qual results remove word 'they' and refer to participants or facilitators when describing who said what.

These have been amended in the appropriate places throughout the manuscript.

The program itself is potentially effective at directing and empowering parent carers to change their own behaviour to improve their health outcomes. The targets for the program seems to be healthy behaviour change impacting health. This needs to be written more directly. The program itself can't improve health. The program promotes behaviour change to help participants achieve better health themselves.

Yes we agree and have added a sentence stating this in the introduction. (Page 4, line 19).

The TiDIER checklist is a very useful checklist to describe all of the aspects of an intervention. This might assist authors to create a summary of the all of the parameters of the intervention--replacing Box 1. The supplementary table doc 1 is detailed and excellent but wondering if you need a better summary in the published article because many readers won't download this?

A TiDIER checklist has been completed and is now supplementary document 1b

In key points--define your homogenous population and state that application to CALD populations requires future consideration and participatory involvement.

We have added text to reflect this in the strengths and limitations key points section on page 4, line 8 and further highlight the point in the discussion that the application of HPC to CALD populations will be explored in a future programme of work. (Page 14, line 28)

In the description of intervention mapping the process is described as if it was iterative and inductive BUT it also seems that the CLANGERS program was applied over the top of this.

Consider whether you were guided by CLANGERS and feedback in the way this is described for consistency.

In our intervention development process we distinguish the underlying concept (CLANGERS), the designing the content of the intervention to be appropriate for parent carers, and tailoring delivery strategies to parent carers to optimise accessibility. So whilst CLANGERS remains foundational, development of content and delivery has been systematic and iterative. We have made amendments to the manuscript when describing intervention mapping in the introduction (page 4, line 29) so as not to conflate the underlying concept and the iterative design process.

If the facilitators were trained--please explain how. What risk management protocols were put in place and how were they included--i.e. carers with mental health issues managed within the group, directing parents to medical assistance, domestic violence issues...other? If this doesn't exist, describe this under future developments because the evidence is that domestic violence is higher and children with disability have a 4 fold risk for abuse and neglect related to parental mental health--another alarming statistic that might be included to justify the crucial nature and need for your program.

Safeguarding procedures have been added to Box 2. (Page 20)

We cannot assume that peer mentors or leaders know what they are doing just because they have lived experience--this aspect needs explaining as the effectiveness is surely related to the skills of the facilitators. i.e. effective groups have been run by health professionals (see systematic review above) but little evidence that peer mentors are as effective in behaviour change. Peer mentors and leaders are effective for support but this intervention aims to change healthy behaviours. Please explain or add more to box 2 (i.e safeguarding and facilitators need more explanation and also who trained facilitators and why they were sufficiently skilled to do so).

Peer mentors are a key component of our intervention to facilitate social identification and peer support processes, but we agree that it is critical that peer facilitators are well-trained and prepared (e.g. regarding safeguarding). Our findings (qualitative and added fidelity of delivery) show that the training of peer facilitators was successful in this trial and this has been highlighted in the discussion (Page 13, line 28).

Details of facilitator recruitment have been added to Box 2. (Page 20)

Please add in research aims or questions clearly with a heading to draw the readers attention

The title of the manuscript has been amended to better reflect the aims of the paper and these have been stated at the end of the introduction (Page 6, line 1).

Methods that you implemented to support the trustworthiness of the qualitative data is needed to show dependability of your qual findings. The authors present the summary of what participants said in their own words and the reader doesn't know how trustworthy this is--i.e. what evidence can you provide which will show that researcher bias what minimised in the analysis of interviews. This is important to explain.

The data to support our summary of key qualitative findings are reported in Boxes 4 and 5, and additional quotes are reported in the Supplementary Document 11 – we have clarified this on page 8, line 4. As we reported in the Methods (Data analysis / qualitative data section), three researchers initially independently coded three transcripts to develop and agree a coding framework, and then half of transcripts were double-coded and the coding of the focus group transcript was checked (we reported this on page 7). The team regularly discussed the analysis and interpretation throughout. Moreover, the data were primarily coded by one researcher but then the findings were written up by another qualitative researcher who again double-checked the coding and analysis during the writing-up process. These steps helped ensure good quality and trustworthiness of the analysis, and minimise any potential researcher bias. We have clarified this on page 7, line 13.

The qual results appeared to be handled deductively with specific feedback sought. Please include a description of the qualitative design and cite the source that drove the qual data analysis process.

In the Methods section (page 7, line 1) we reported that we took both an inductive and deductive approach to thematic analysis. We initially took an inductive approach to coding the transcripts in order to capture experiences and views discussed by the participants/facilitators. Based on this, and the independent coding, we then agreed on a coding framework. As we reported on page 7: 'While the detailed codes were developed inductively, the categories were more deductive and followed the key areas of interest in the process evaluation, reflecting the interview topic guide.' Thus, the process evaluation questions influenced our deductive approach. We hope this clarification is satisfactory to the Reviewer.

Reformat the methods and results sections so that they are sequential and not blended as they currently are. This is confusing for the reader and detracts from the rigour that you applied to the research. For example, under qual study--describe methods and then results for who and how many applied etc.

We report the qualitative and quantitative data together in the findings section under each theme (a usual practice in mixed methods research) so that the reader can understand the whole rather than the sum of the individual qualitative and quantitative parts. This provides a strong and coherent narrative for linking findings to subsequent intervention development and trial outcomes.

Who interview participants and how did you control for positive reporting if interviews were not anonymised. This applies to the quant data too. Were participants able to be anonymous and honest or is there a risk that participants were positively reporting.

In the Methods section under data collection (page 6, line 26), we have added who carried out the qualitative data collection and that these researchers were not directly involved in delivering the programme. This information can be also found in Box 3.

Selective sample is a limitation for all studies. We could not sample participants for interviews based on survey responses as these were anonymous. However, as reported in Box 3, interviewees *'were sampled to ensure that two out of the four male carers in the study (1 control and 1 intervention) were interviewed and the range of parent carer challenges were represented (selection was based on Lead Facilitator comments and participant end of programme feedback)'*. We have clarified how we sampled participants for interviews and facilitators for the focus group in the Methods (page 6, line 19) and in Box 3.

As explained above and in the paper, we aimed to collect diverse experiences and views, and reassured participants and facilitators that they can be as honest as they wish and that we welcomed critical or negative comments to improve the programme in the future. Indeed, we did not get only positive reporting (neither in qualitative nor quantitative data). We reported some negative experiences in the Findings (e.g. groups being too small, content of the online videos not liked etc.) and these experiences and all suggestions were used in the intervention refinement process. Changes made to the training, delivery and content of the programme showed that less positive feedback from all data sources was utilised to refine the intervention.

Add in all ethical approvals and considerations--the study is about a vulnerable group but ethics is not explained--please add this to the manuscript.

Ethical approval was granted for this study and an additional sentence about safeguarding has been added (page 16, line 9).

The research team are to be congratulated on an interesting and thorough study that has obviously included co-design and co-delivery with the target group.

Thank you

The 14 participants reporting neutral or not satisfied with the group were potentially a rich source of feedback to make changes to improve the program. Please expand what was learned from these questionnaires or interviews.

We agree with the Reviewer. As clarified above and reported on page 12, line 11: 'less positive experiences and suggestions were carefully considered and used in the refinement process (Figure 2)'.

What did participants think of the instruments and data collection etc --i.e. feasibility of research design as well as the content and delivery of the program. This seemed to be the intention of the study but results/ explanation may need to be added.

These data are reported in the partner paper which has been submitted and it is not the aim of this paper to report feasibility and acceptability of trial processes.

The impact on this small group also relates to safeguarding and the skill of facilitators. Please show how harm can be minimised in future programs from these learnings. You say "however less positive experiences were carefully considered in the refinement process" but please explain how. Please add some negative but constructive quote from participants into box 5 to represent these experiences (if possible)

All refinements reported are based on the less positive experiences/views and suggestions, as reported in the qualitative findings. We have added small clarifications in the Refinements section to make the links more explicit (pages 12-13), while trying to avoid extending the length of the manuscript considerably. We have also added less positive quotes in box 5.

Discussion is excellent and brings paper together for the reader.

Thank you

Supp doc 11 is extensive but worth including.

Thank you

More cross referencing to all supp docs, tables needed.

All supplementary documents have been cross referenced in the manuscript, with additional references included when appropriate.

VERSION 2 – REVIEW

REVIEWER	Jessica Ward Children's Hospital of Los Angeles
REVIEW RETURNED	03-May-2021
GENERAL COMMENTS	The clarity of the manuscript is much improved. Thank you for diligently addressing the reviewer comments.
REVIEWER	Helen Bourke-Taylor Monash University Faculty of Medicine Nursing and Health Sciences, School of Primary and Allied Health Care
REVIEW RETURNED	12-May-2021
GENERAL COMMENTS	Thank you for responding to the issues raised in the first review. I am unsure why you were unable to access the PDF document that I completed for the review. I didn't keep a copy although the journal

	has the upload. However, i think most of my comments have been attended too.  1. The TiDIER table is very useful at directing the reader to the sections of the manuscript--well done. 2. I understand that you did not set out to make it a program for mothers and this is understandable but having said that women's health issues are different to men's health issues and I wonder if the discussion should acknowledge this? For example, women have many more screens available to them and help seeking and service use are a big factor in delayed diagnosis for carers. It may be that CLANGERS is more generic than that? Up to you but women's health might have some mention please. Also accommodations in future programs? Or reasons why you are or are not doing this. Further discussion about these issues and of better including fathers might also occur, because the 'primary carer' inclusion criteria is somewhat misleading --see recent article for reasons why: Bourke-Taylor, H.M., Cotter, C., Joyce, K., Morgan, P., Reddihough, D. Brown, T. (2021) Fathers of children with a disability: Health, work and family life issues. Disability and Rehabilitation (Early View, 2021) Please add something about the gender/role/primary carer what happened in the project and what you recommend to both address women's health for the expected higher proportion of women and also how and why fathers need to be engaged.  3. recommendations were made about removing words like 'felt' etc. However I cannot see any changes in the R1. Can you please highlight these or direct me to then? 4. Satisfactory description of qual data analysis although trustworthiness (credibility, dependability, replicability...) is not addressed. How did you ensure trustworthiness--please provide examples. This is important to manage the positive reporting aspect of your results, even though you did respond to this question okay in the response to reviewers. Please expand on trustworthiness. 6. Please describe your purposive sampling as maximum variation sampling as well as you sought diverse feedback--this could be explained briefly for the readers benefit. It is not called 'selective sampling' it is 'purposive maximum variation sampling' in qual research. 7. thanks for including CALD groups in discussion. However you refer to the program as a parenting intervention which it is not. please correct this. i.e. "However, adapting parenting interventions for different..." Great work. Thanks for the opportunity to read and comment on your work.
--	--

VERSION 2 – AUTHOR RESPONSE

Reviewer: 1

Dr. Jessica Ward, Children's Hospital of Los Angeles Comments to the Author:

The clarity of the manuscript is much improved. Thank you for diligently addressing the reviewer comments.

Thank you

Reviewer: 2

Dr. Helen Bourke-Taylor, Monash University Faculty of Medicine Nursing and Health Sciences Comments to the Author:

Thank you for responding to the issues raised in the first review. I am unsure why you were unable to access the PDF document that I completed for the review. I didn't keep a copy although the journal has the upload. However, i think most of my comments have been attended too.

1. *The TiDIER table is very useful at directing the reader to the sections of the manuscript--well done.*

Thank you

2. *I understand that you did not set out to make it a program for mothers and this is understandable but having said that women's health issues are different to men's health issues and I wonder if the discussion should acknowledge this? For example, women have many more screens available to them and help seeking and service use are a big factor in delayed diagnosis for carers. It may be that CLANGERS is more generic than that? Up to you but women's health might have some mention please. Also accommodations in future programs? Or reasons why you are or are not doing this. Further discussion about these issues and of better including fathers might also occur, because the 'primary carer' inclusion criteria is somewhat misleading --see recent article for reasons why: Bourke-Taylor, H.M., Cotter, C., Joyce, K., Morgan, P., Reddihough, D. Brown, T. (2021) Fathers of children with a disability: Health, work and family life issues. Disability and Rehabilitation (Early View, 2021)*

Please add something about the gender/role/primary carer what happened in the project and what you recommend to both address women's health for the expected higher proportion of women and also how and why fathers need to be engaged.

Our focus has always been on the promotion of general health and wellbeing for primary carers and not on 'women's health' per se. Other programmes may aim to address these, however the HPC programme was designed to be more generic. Both men and women were involved in developing the programme. Although we appreciate that women (who represent most primary carers), have unique health needs, we do not feel comfortable making recommendations outside of our expertise.

We have had a few men in our groups and their experience has been positive. It would appear the social identity as parent carers supersedes to a certain extent sex/gender. Future work might explore how best to engage/support fathers and other subgroups of carers (cultural/ethnic) and evaluate specific HPC groups. Your paper will be extremely useful in helping us to develop this piece of work.

We have added a comment about the gender bias of participants in the discussion (page 14).

3. *recommendations were made about removing words like 'felt' etc. However I cannot see any changes in the R1. Can you please highlight these or direct me to them?*

There were 4 uses of the term 'felt' by the authors and all have been amended. The amendments can be found in the following places

Results section of abstract

Page 10, line 18

Page 14, line 1 and 4.

4. *Satisfactory description of qual data analysis although trustworthiness (credibility, dependability, replicability...) is not addressed. How did you ensure trustworthiness--please provide examples. This is important to manage the positive reporting aspect of your results, even though you did respond to this question okay in the response to reviewers. Please expand on trustworthiness.*

The quality and trustworthiness of the qualitative analysis was ensured through our sampling methods (maximum variation sampling), qualitative data collection methods (researchers not involved in delivering the intervention collected qualitative data), data analysis (three researchers independently coded transcripts to develop the coding framework, 50% of transcripts were double coded, coding was cross checked and there were regular team meetings to discuss analysis and

interpretation). Qualitative findings from the parent interviews were triangulated with the data arising from the focus group and the questionnaires.

The last sentence from the paragraph above has been added to the Data Analysis section (page 7)

5. *Please describe your purposive sampling as maximum variation sampling as well as you sought diverse feedback--this could be explained briefly for the readers benefit. It is not called 'selective sampling' it is 'purposive maximum variation sampling' in qual research.*

We have amended to 'participants were purposely sampled for maximum variation, to include male and female perspectives, a range of engagement levels, and parent carer challenges' (Methods page 6, line 19).

6. *Thanks for including CALD groups in discussion. However you refer to the program as a parenting intervention which it is not. please correct this. i.e. "However, adapting parenting interventions for different..."*

We have deleted the term parenting (Page 14, line 31)

Great work. Thanks for the opportunity to read and comment on your work.

Thank you. The main feasibility findings have just been published (link below)

<https://pilotfeasibilitystudies.biomedcentral.com/articles/10.1186/s40814-021-00881-5>